# Estimation of Average Speed of Road Vehicles by Sound Intensity Analysis

**DOI:** 10.3390/s21165337

**Published:** 2021-08-07

**Authors:** Józef Kotus, Grzegorz Szwoch

**Affiliations:** Department of Multimedia Systems, Faculty of Electronics, Telecommunication and Informatics, Gdańsk University of Technology, 80-233 Gdańsk, Poland; grzegorz.szwoch@pg.edu.pl

**Keywords:** traffic analysis, speed estimation, sound intensity, acoustic vector sensor

## Abstract

Constant monitoring of road traffic is important part of modern smart city systems. The proposed method estimates average speed of road vehicles in the observation period, using a passive acoustic vector sensor. Speed estimation based on sound intensity analysis is a novel approach to the described problem. Sound intensity in two orthogonal axes is measured with a sensor placed alongside the road. Position of the apparent sound source when a vehicle passes by the sensor is estimated by means of sound intensity analysis in three frequency bands: 1 kHz, 2 kHz and 4 kHz. The position signals calculated for each vehicle are averaged in the analysis time frames, and the average speed estimate is calculated using a linear regression. The proposed method was validated in two experiments, one with controlled vehicle speed and another with real, unrestricted traffic. The calculated speed estimates were compared with the reference lidar and radar sensors. Average estimation error from all experiment was 1.4% and the maximum error was 3.2%. The results confirm that the proposed method allow for estimation of time-averaged road traffic speed with accuracy sufficient for gathering traffic statistics, e.g., in a smart city monitoring station.

## 1. Introduction

Intensity of road traffic is increasing every year. Therefore, efficient means of traffic monitoring are needed by both the vehicle drivers and the road network managers. Drivers need up-to-date information on traffic congestion in order to choose the quickest route. Currently, vehicle navigation systems rely mainly on dynamic data collected from their users. Authorities that manage the road networks collect data from radar sensors and inductive loops, and from ad-hoc measurements. Smart city solutions are the modern approach to the management of urban areas [1]. Traffic monitoring in a smart city system requires installing a large number of efficient and cost-effective sensors. A network of monitoring stations collects data (from various sensors) that is analyzed in the cloud. The results are available in real-time for drivers, the authorities, and the automated intelligent transportation systems. The monitoring stations often measure traffic speed averaged in time slots [2].

Currently, state-of-art traffic monitoring sensors include radars, lidars and inductive loops [3,4,5]. Newer trends include video analysis from cameras [6,7] and data collection from vehicles [8]. Radars and lidars are used whenever an accurate measurement of vehicles speed is required to ensure safety of drivers and pedestrians (e.g., for traffic law enforcement) [9]. Such measurement devices must be certified; therefore, they are expensive, and they are used mainly for short-term measurements. Cheaper radar sensors, with limited accuracy, are often installed on urban roads, e.g., at the pedestrian crossings. However, radar sensors are active devices, emitting electromagnetic waves. A large number of radar sensors installed in the monitored area increases environmental pollution with electromagnetic waves that may interfere with cellular networks, Wi-Fi devices, etc. The radar itself is susceptible for electromagnetic interference from power lines, airport radar systems, etc., which makes traffic monitoring in certain locations problematic or even impossible. Also, adverse weather conditions, such as intense rainfall, prevent radar sensors from working correctly. Current urban traffic monitoring systems are often based on inductive loops. While they are effective, they need to be installed in the road surface, which may be problematic in some cases. Other means of traffic monitoring, such as pneumatic tubes, are only suitable for short-term measurements, as they obstruct the traffic. Smart city traffic monitoring systems, covering large areas, should preferably consist of passive sensors, robust to interference from the environment. It should be possible to install the sensors in convenient places, without interfering with the existing road infrastructure. Finally, the cost of the sensor (both the device itself and its power consumption) should be low.

Acoustic sensors seem to be underutilized in traffic monitoring systems. They are passive sensors, relying only on the analysis of environmental sounds. Passing vehicles emit sounds from various parts: the engine, the exhaust, the tires, etc. Even if the electric cars become the majority of the road vehicles, their tires still produce noise as a result of contact with the road surface, and that sound can be picked up and analyzed. A single microphone only allows for pressure analysis; acoustic events may be detected, but it is problematic to determine whether this event is related to a road vehicle. The level of noise in urban areas is often high, so the sensor must be able to not only detect the sound, but also determine its direction. Most devices that perform detection of the incoming sound direction are large microphone arrays that are not practical for installation in traffic monitoring stations.

The aim of the research presented in this paper was to propose an alternative method of estimating average traffic speed. We base our algorithm on the analysis of sound intensity measured by an acoustic sensor. The algorithm is intended to provide an average speed of road vehicles measured in defined time windows. The requirement is that the accuracy of the proposed method is comparable with the state-of-art methods, such as radars, but it has advantages over standard sensors: it is a passive sensor, it is not susceptible to electromagnetic interference, it can be easily installed at a chosen location and it has low cost (of construction and usage), so that it may be implemented e.g., in a large smart city system.

An acoustic vector sensor (AVS) is a device able to analyze both the intensity and the direction of sound waves, using a setup of sensors (e.g., microphones) contained in a small enclosure [10,11]. An AVS, such as the one used in the experiments described in this paper, may be constructed from low-cost microphones, and the sensor size is small. Therefore, such sensors are suitable for the smart city monitoring stations. In our earlier research, we employed an AVS for detection of acoustic events and determining direction of the incoming sounds [12]. We successfully applied an AVS for determining traffic intensity, i.e., providing a reliable data on the number of vehicles within the observation period [13,14]. An advantage of the AVS, compared with a single microphone, is the ability to determine the incoming sound direction, which allows for establishing the direction of a vehicle movement, and for filtering out acoustic events not related to road vehicles. In this paper we extend this approach, and we propose a novel method of estimating the average speed of road traffic, with a sufficient accuracy for the task of traffic monitoring in a distributed smart city system.

Analysis of audio signals, and specifically sound intensity signals, is rarely employed for traffic monitoring. Most of the published works on the analysis of sounds produced by road vehicles is related to traffic noise measurement and prediction, e.g., the Harmonoise project [15]. An interesting work, from the point of view of the method described in this paper, was published by Ballesteros et al. [16]. They evaluated spatial distribution of noise emitted by a car, in frequency bands, using a large beamforming array. Their conclusions regarding spatial and spectral distribution of sound sources in a passenger car were utilized in our research.

In the context of acoustic traffic monitoring, the published works usually employ large microphone arrays. For example, Na et al. [17] used an array consisting of 37 microphones for detection of vehicle positions on multiple lanes. Barbagli et al. [18] used a wireless sensors network for estimation of traffic intensity. Chen et al. [19] applied correlation-based sound field mapping to signals from a microphone array. An alternative approach is to use two microphones positioned alongside the road. Duffner et al. [20] used a two-microphone setup and cross-power spectrum algorithm for detection of road vehicles. Although they mention speed estimation, no experimental data were provided. López-Valcarce et al. [21] employed a similar setup and a maximum likelihood algorithm for estimation of a vehicle speed from its acoustic signature. Cevher et al. [22] used acoustic wave patterns obtained with a single microphone for vehicle speed estimation based on vehicle profile vectors. Ishida et al. [23] employed a time-difference sound mapping technique based on dynamic time warping for vehicle counting. Other works focus on estimation of traffic intensity based on audio analysis. Warghade and Deshpande [24] used a single omnidirectional microphone to evaluate traffic intensity on a three-degree scale. Gatto and Forster [25] applied machine learning to detect traffic congestion using a single microphone. An interesting approach proposed by Vij and Aggarwal [26] relies on crowdsourced sound acquisition from smartphone users to detect traffic state.

In the publications listed above, acoustic traffic monitoring was based on analysis of sound pressure, using complex and computationally expensive algorithms. Contrary to that, a novel approach presented in this paper focuses on analysis of sound intensity. The algorithm is simple and suitable for implementation on low-cost processors. The remaining part of the paper is organized as follows. First, we describe a method of calculating sound intensity signals from pressure signals obtained from the sensor and calculating the source position from the intensity signals. Next, we describe a model of an ideal, moving point source observed by the sensor and we compare it with the results obtained from the actual vehicle pass. In the following Section, we present the proposed method of estimating the average speed of road vehicles within an observation window. Next, we present the results of experiments that validated the proposed method, and the paper ends with Conclusion.

## 2. Materials and Methods

### 2.1. Intensity Signals and Source Direction

Sound intensity is a measure that describes the energy flow in sound waves, defined as the power carried by sound waves per unit area in a direction perpendicular to that area [27,28]. Instantaneous sound intensity is calculated as a product of sound pressure *p* and particle velocity **u**. The velocity **u** is a vector, direction of which corresponds to the sound wave direction. Therefore, the intensity is also a vector. In practice, sound intensity **I** is averaged in time windows *T*:(1)$$I=\frac{1}{T}\int_{0}^{T}p(t)u(t)\;dt$$

An acoustic vector sensor (AVS), also called a sound intensity probe, measures the magnitude *I* of the sound intensity [10]. A single-dimensional (1D) AVS may be constructed from one pressure sensor and one velocity sensor (a p-u probe) [11], or from two closely spaced pressure sensors (a p-p probe) [12]. In the latter, pressure *p* at the middle point between the sensors is calculated as an average of both sensors pressure values (*p*_1_, *p*_2_), and the magnitude *u* of the velocity vector is computed as the integral of a pressure gradient:(2)$$p\left(t\right)=\frac{p_{1}\left(t\right)+p_{2}\left(t\right)}{2}$$
(3)$$u\left(t\right)=\int_{-\infty }^{t}\left(p_{2}\left(t\right)-p_{1}\left(t\right)\right)\;dt$$

The direction of the velocity vector is determined by the axis from *p*_1_ to *p*_2_. Averaged intensity *I* is calculated by time-averaging the product of pressure and velocity. The sensor must be calibrated if the measured intensity value is to have a physical meaning.

Placing two 1D intensity sensors on orthogonal axes so that the middle points of both sensors are in the same position, creates a two-dimensional (2D) AVS. The axes of a 2D AVS form a *XY* plane, and the azimuth *ϕ* of the incoming sound may be calculated as:(4)$$\phi =\arctan\left(\frac{I_{Y}}{I_{X}}\right)$$
where *I_X_*, *I_Y_* are intensity values (magnitudes of the intensity vectors) measured in axes X and Y, respectively.

Similarly, a third axis Z, orthogonal to the XY plane, may be added in a way that the middle points of all three sensors are in the same position. A 3D AVS allows for measuring both the azimuth *ϕ* and the elevation *θ*, given by:(5)$$\theta =\arctan\left(\frac{I_{Z}}{\sqrt{I_{X}^{2}+I_{Y}^{2}}}\right)$$
where *I_Z_* is the magnitude of intensity vector measured on the *Z* axis.

In a three-dimensional space, a 2D AVS determines a plane, on which the sound source is positioned, a 3D AVS determines a ray that originates from the sensor and intersects the sound source. An AVS is not able to determine the distance to sound source, so it cannot find the exact position of the source in polar coordinates.

Obtaining accurate azimuth and elevation values from the AVS requires that all pressure signals are aligned in amplitude, and the velocity signals are aligned in phase. This is ensured by means of the calibration procedure that calculates the amplitude and phase correction functions [29]. These functions are applied during the intensity measurement.

### 2.2. Analysis of an Ideal Moving Point Source

In this example, the calculated intensity and angles are used to track an ideal, omnidirectional point source emitting acoustic energy with a constant power, moving along a linear path with a constant speed *v*. The *Z* axis is omitted for simplicity. The sensor is oriented so that its *X* axis is orthogonal to the source trajectory, the *Y* axis is parallel to the trajectory, the distance *x* between the *X* axis and the source trajectory is constant (Figure 1). The position of the sound source is (*x*, *y*). Assuming that the source emits constant power *P* and the intensity is inversely proportional to the squared distance from the source, intensity observed by the sensor is given by:(6)$$I_{X}\sim P\frac{x}{{\left(x^{2}+y^{2}\right)}^{3/2}}$$
(7)$$I_{Y}\sim P\frac{y}{{\left(x^{2}+y^{2}\right)}^{3/2}}$$

Position *y* of the source may be calculated from the intensity measured in *X* and *Y* axes:(8)$$y\left(t\right)=x\cdot \tan\left(\phi \left(t\right)\right)=x\cdot \tan\left(\arctan\left(\frac{I_{Y}\left(t\right)}{I_{X}\left(t\right)}\right)\right)=x\cdot \frac{I_{Y}\left(t\right)}{I_{X}\left(t\right)}$$

Since the position *y* changes in time, we will be using the term ‘position signal’ throughout the paper to describe *y*(*t*).

Speed *v* of the source may be calculated as:(9)$$v\left(t\right)=\frac{dy\left(t\right)}{dt}=x\cdot \frac{d}{dt}\frac{I_{Y}\left(t\right)}{I_{X}\left(t\right)}$$

For velocity estimation with this method, it is essential that the accurate value of *x* is known. However, *x* cannot be measured directly with an AVS, it can only be estimated. A possible solution to this problem is presented further in the paper.

### 2.3. Analysis of a Single Vehicle Pass

The aim of the presented method is to track a moving vehicle using a sensor placed alongside the road and oriented as in Figure 1. However, a real road vehicle cannot be represented as an ideal point source because it is a superposition of multiple sound sources (the engine, the exhaust, the tires, the vehicle body, etc.). These sources differ in intensity and directivity, and their intensity may change with time and depend on frequency. As a result, a road vehicle is a complex setup of individual sound sources, also the distances between these sources are comparable with the distance between the sensor and the vehicle. Moreover, each vehicle is a distinct setup of sound sources. Speed of a vehicle may also change while the vehicle passes the sensor. Azimuth and elevation obtained from the sensor indicate an apparent point source that is a superposition of the individual sources. Intensity, directivity, and position of that source inside the vehicle are time-dependent.

Figure 2 presents the intensity in *X*-*Y* directions, the azimuth, and the estimated position, computed from recorded signals of a passenger car, passing by the sensor at an approximately constant speed of 66.9 km/h (18.6 m/s), ca. 3.8 m from the sensor, from left to right. These plots are compared with an ideal point source moving at the same speed and distance from the sensor. Differences are clearly visible. Sound intensity plot in the perpendicular direction (*I_X_*) for a real vehicle is shifted back in time (precedes the point source) and the plot of intensity in parallel direction (*I_Y_*) is asymmetrical, with more weight on the left of the zero-crossing point. This indicates that the vehicle propagates most of the sound energy towards the front of the vehicle, mainly due to a horn effect [30]. The azimuth plot of the vehicle also deviates from the point source, and the position plot shows even larger differences compared with a linear plot of the point source, for the reasons described earlier in this Section. It should be also noted that when the vehicle is far away from the sensor, signal-to-noise ratio becomes very low and the azimuth and the position are not measurable. Therefore, only a short section of the signal (near the zero azimuth point) is usable for analysis.

To conclude, in order to measure speed of each individual vehicle, its exact setup of individual sound sources would have to be known. This is not realizable with the presented sensor, it would require employing a large sensor array. However, the aim of the method presented here is to estimate an average speed of vehicles within the observation period, by processing a set of position signals obtained from multiple vehicles. The details of this method are presented in the subsequent sections of the paper.

### 2.4. Sound Intensity in Frequency Bands

Estimation of a vehicle speed with an AVS requires that a fixed point within the vehicle is tracked as the vehicle passes by the sensor. However, the horizontal and vertical position of the apparent sound source, observed by the sensor, depends on both the vehicle position and the frequency. Therefore, it is reasonable to perform the analysis of sound intensity in frequency bands. Ballesteros et al. published a study on spatial distribution of noise from a passing vehicle, in third-octave frequency bands [16]. At 1 kHz, the whole vehicle emits noise, with a car engine being the dominant source. The apparent source is positioned approximately in the middle of the vehicle, shifted towards the engine and towards the tires that are closer to the sensor. At 2 kHz, the influence of the engine decreases and the tire noise becomes more prominent. The apparent source is expected to be positioned lower than for 1 kHz. Finally, at 4 kHz, tire noise becomes the main component. The apparent source is positioned close to the ground, but its horizontal position, resulting from superposition of four separate sources, depends on the horizontal position (azimuth) of the vehicle, relative to the sensor. It should also be noted that noise intensity decreases with frequency, so the sound intensity measured in the 1 kHz frequency band is usually significantly higher than in the 4 kHz band. At the same time, attenuation of sound intensity with increasing distance between the source and the sensor depends on the frequency. Therefore, relationship between intensity measured in 1 kHz and 4 kHz bands also depends on the distance of the vehicle from the sensor.

Based on the observations described above, we decided to perform the analysis of sound intensity in three octave bands centered at frequencies: 1 kHz, 2 kHz, and 4 kHz. Lower frequency bands were rejected because they usually contain a high level of noise (e.g., wind), and higher frequency bands have too low signal-to-noise ratio [14,15,16]. The task is to track a point that remains at a relatively constant position within a vehicle during the measurement. In order to determine the position *y* of the source, we discard the 4 kHz band, because the intensity from four separate sources (tires) causes the apparent sound source to move as the azimuth increases. Either of the 1 kHz and 2 kHz bands (or both) can be used for determining the source position. We decided to perform an independent analysis of sound intensity in two bands and to average the results, as a form of result smoothing, reducing the measurement errors.

Determining the distance *x* between the source trajectory and the sensor, which is required for speed estimation according to Equation (8), is more problematic. A vertical position of the sound source is not known, and it cannot be measured. However, since the height of the sensor above the ground can be measured, it is possible to calculate the distance between the sensor and a projection of the apparent sound source on the ground, using simple trigonometric relations (Figure 1b). Consider the case when the source azimuth is zero. For the 1 kHz band, the source is positioned approximately at the middle point of the vehicle, so its projection to the ground level is usually beyond the vehicle. For the 4 kHz band, the source projection is positioned close to the intersection of lines connecting the car tires. In the proposed method, the values calculated for the 1 kHz and 4 kHz bands are averaged:(10)$$x=\frac{h_{s}}{2}\left(\cot\theta_{1k}+\cot\theta_{4k}\right)$$
where *h_s_* is the sensor height above the ground. In order to reduce the measurement error, values of *x* are averaged over the time window of 0.2 s, centered at the point of zero azimuth.

It is our assumption that with this approach, we obtain a distance to the point situated near the tires further from the sensor, approximately at the half of distance between the front and the rear tire, and that this point remains constant relative to the vehicle when the source is observed within the azimuth range used for the measurements (about −45 to 45 degrees). In order to verify this assumption, we performed the following experiment. Several passes of vehicles moving at a constant speed were recorded with the AVS. A certified lidar-based device (Vitronic Poliscan) was used to measure the vehicle speed and to determine the position of two edge points at the front of vehicle’s body, called a near and a far reference point. Based on these measurements, movement of two reference points was estimated using a point source model presented earlier and compared with the sound intensity signals computed for the vehicle, in three frequency bands. The obtained azimuth and elevation signals for a single vehicle are shown in Figure 3 and Figure 4. The results of azimuth measurement in the 1 kHz and 2 kHz bands mostly overlap with each other and with the modelled far reference point. The 4 kHz band values deviate from the other bands as the azimuth increases, as expected. From the elevation plot it may be observed that at the zero azimuth, the 4 kHz value indicates a point consistent with the near reference point, while the 1 kHz point is well beyond the vehicle (larger negative elevation means that the point is closer to the sensor). However, for larger absolute azimuth values, the source projection in the 1 kHz band moves closer to the far reference, while the 4 kHz band values indicate a position before the vehicle. The values in the 2 kHz band are inconclusive. Analysis of elevation is required only in a short time window around the zero-azimuth point (±0.1 s), and the average of results from the 1 kHz and 4 kHz bands follows the modelled movement of the far reference point within this time frame. Therefore, the assumption stated earlier is confirmed for the far reference point as the tracked source.

### 2.5. Source Position for a Single Vehicle

The method presented here computes the position signal for a single vehicle. Procedure for calculation of intensity signals from pressure signals provided by the AVS is presented in Figure 5. In this paper, we assume that the pressure signals are discrete and uniformly sampled. The pre-processing stage performs amplitude and phase equalization of the pressure signals to minimize differences between the transfer functions of microphones that would contribute to errors in sound intensity measurements [29]. Next, signal sections that represent individual vehicles are determined by analysis of the intensity and azimuth functions [14]. For each vehicle, sound intensity in three axes, as well as azimuth and elevation functions, are calculated in three octave bands (1 kHz, 2 kHz, and 4 kHz). The results are then used to calculate the position of the vehicle, using the previously presented method.

The source position is calculated from the intensity signals (Equation (8)). However, in practical situations, sound intensity measured with the AVS is contaminated by noise, resulting from sensor imperfections, sound source movement, external factors (e.g., wind noise), etc. Therefore, Equation (8) in practice becomes:(11)$$y\left(t\right)=x\cdot \frac{I_{Y}\left(t\right)+\eta_{X}\left(\mu_{X},\sigma_{X}\right)}{I_{X}\left(t\right)+\eta_{Y}\left(\mu_{Y},\sigma_{Y}\right)}$$
where *η* is Gaussian noise with mean *μ* and standard deviation *σ*. As a result, position estimation from noisy sound intensity signals is inaccurate.

Suppression of noise involves smoothing the intensity signals obtained for a single vehicle, in each frequency band. This is necessary to reduce the influence of noise on calculation of the source angles, distance, and position. Several methods of intensity smoothing were examined, the method that provided optimal results performs filtering the intensity signal with a rolling median filter with time window of ca. 267 ms, followed by a rolling average filter with averaging time of ca. 139 ms.

The smoothed intensity signals are used to compute azimuth, elevation, distance, and position of the source on the road. Distance *x* to the source is estimated using the method described earlier, as shown in Figure 6. Position *y* is calculated from Equation (8), using the estimated distance *x* (Equation (10)), within a time window of 0.4 s, centered at the zero-azimuth point. The position signals are calculated independently in the 1 kHz and 2 kHz bands, and the results are averaged in order to diminish the influence of noise (Figure 7).

### 2.6. Estimation of Time-Averaged Traffic Speed

Accuracy of determining the distance *x* from the sensor to the path of the moving source is the main factor limiting the accuracy of speed estimation. Despite the noise suppression, the distance estimates are still noisy, and small changes of distance *x* cause large errors in speed estimation. However, it is expected that errors in distance estimation (and speed estimation) observed for a large number of vehicles form a normal distribution (this was confirmed in the experiments described further in the paper). Therefore, we assumed that if a sufficiently large number of position signals from individual vehicles are averaged over the observation period, the influence of the estimation noise diminishes, and the speed estimate computed from the average position signal will be close to the actual time-averaged traffic speed. This assumption was verified during the experiments and the results are presented in the paper.

A method of estimation of the averaged traffic speed from position signals is shown in Figure 8. Instead of simply averaging the estimated speed values of individual vehicles, the proposed approach calculates an average position signal from all vehicles within the observation period. The aim is to reduce the measurement noise present in individual position signals. First, the position signals *y* from all analyzed vehicles are synchronized so that their zero-azimuth points are aligned. The position signals (discrete, with an identical sampling period) are resampled using a linear interpolation method, so that the discrete values of all signals occur at the same time instants, and one in each signal represents exactly the zero-azimuth point.

In the next step, *M* time-aligned position signals *y_i_* are averaged over time, resulting in a position signal *y_a_* of the ‘averaged’ vehicle:(12)$$y_{a,n}=\frac{1}{M}\sum_{i=1}^{M}y_{i,n}$$

Speed estimation is obtained by performing a linear regression on the averaged position signal which consists of *N* discrete-time points (*t_n_*, *y_a,n_*). The regression coefficient is given by:(13)$$v_{est}={\bar{y}}_{a}-\frac{\sum_{n=0}^{N-1}\left(t_{n}-\bar{t}\right)\left(y_{a,n}-{\bar{y}}_{a}\right)}{\sum_{n=0}^{N-1}{\left(t_{n}-\bar{t}\right)}^{2}}\bar{t}$$
where the x-bar indicates mean values. The position values are evenly spaced, so *t_n_* = *n*·*T*, where *T* is the sampling period of the position signal in seconds. Therefore:(14)$$\bar{t}=\frac{1}{N}\sum_{n=0}^{N-1}nT=\frac{N-1}{2}T$$

The denominator in Equation (13) is:(15)$$\sum_{n=0}^{N-1}{\left(t_{n}-\bar{t}\right)}^{2}=\sum_{n=0}^{N-1}{\left(nT-\frac{N-1}{2}T\right)}^{2}=\frac{\left(N^{3}-N\right)\;T^{2}}{12}$$

Substituting Equations (14) and (15) and the means of position values into Equation (13), we obtain the regression coefficient which is the calculated speed estimate expressed in meters per second:(16)$$v_{est}=\frac{1}{N}\sum_{n=0}^{N-1}y_{a,n}-\frac{6\cdot \left(N-1\right)}{\left(N^{3}-N\right)\;T}\sum_{n=0}^{N-1}\left[\left(y_{a,n}-\frac{1}{N}\sum_{m=0}^{N-1}y_{a,m}\right)\;\left(\left(n-\frac{N-1}{2}\right)\;T\right)\right]\;$$

## 3. Results and Discussion

For the purpose of validation of the proposed method, a custom AVS device was constructed from six digital, omnidirectional MEMS microphones (InvenSense INMP441, sensitivity −26 dBFS [31]). The microphones were mounted in a cube, distance between microphones on each axis was ca. 0.01 m (Figure 9a), forming a three-dimensional AVS working on a p-p principle [32]. Spacing between the microphones allows for sound intensity analysis up to ca. 10 kHz. The pressure signals from all microphones, sampled at 48 kHz, with 24-bit resolution, were passed through an I2S-USB interface to a microcomputer or a portable computer and recorded on a disk for an offline analysis. An online analysis on a Raspberry Pi 3 microcomputer was also tested and it worked correctly.

The correction functions were obtained by means of the calibration procedure, performed in an anechoic chamber. One amplitude correction function for each pressure signal (each microphone) and one phase correction function for each axis (each acoustic velocity signal) were calculated [29]. The correction was applied to the signals during the calculations, using finite impulse response digital filters of length 512. Instantaneous sound intensity on each axis was calculated using Equations (1)–(3), and then averaged in non-overlapping windows of 256 samples. The resulting time-averaged intensity signals were sampled uniformly at 187.5 Hz (temporal resolution 5.33 ms).

The input data to the proposed algorithm for speed estimation are sufficiently long signal sections representing moving vehicles, around the zero azimuth point. Therefore, intensity signals recorded by the sensor need to be processed by a vehicle detector. Such a detector is not the part of the presented algorithm which does not rely on any specific detection method. In the experiments, we used a detector described in our previous publication [14]. The detection is based on two criteria: presence of a peak in *I_X_* and presence of a smooth transition in the azimuth signal, with zero crossing. If these two criteria are fulfilled, the zero-azimuth point is found and the analysis window of ±1 s is cut around it for further analysis.

Two experiments were performed. The first one (experiment M) was conducted in controlled conditions. Four passenger cars were recorded, and the drivers were asked to drive at a constant speed from the range 70 km/h to 100 km/h. A total of 31 runs were recorded. The reference (state-of-art) sensor was a lidar-based device (Vitronic Poliscan [33]) which was used to measure the actual vehicle speed (with resolution of 1 km/h), as well as the vehicle position on the road, which was used to verify the method of distance estimation, described earlier. We also tried to use a Doppler radar as a second reference device, but high level of external electromagnetic interference (a strong harmonic signal that was added to the signals reflected from the vehicles) made the measurements impossible. An average speed of all 31 runs was 79.5 km/h (22.1 m/s). The road section at the outskirts of a town was straight, the surface was dry and the traffic was very low, so the measured vehicles were isolated. The AVS was placed on a tripod, about 2.6 m from the road edge, at a height of 1.7 m (Figure 9b), oriented as in Figure 1. The results from this experiment were used for the initial validation and tuning of the method.

The second experiment was performed in an uncontrolled environment, to validate the method in a real-life scenario. The measurements were performed on a rural road in the settlement, on a straight road section (one lane in each direction), in dry conditions. The sensor was mounted in an enclosure placed 4 m away from the road edge, 3.2 m above the ground, and it was recording all vehicles moving through the observed section of the road in an unobstructed road traffic. Recordings of isolated vehicles made during the night hours were selected for the calculation. Four one-hour long time slots (experiments L1, L2, L3, and L4) were analyzed, with 8, 7, 5 and 7 observed vehicles in each slot, respectively. We aim to show that a small number of vehicles is sufficient to obtain an accurate estimate of average vehicle speed in the observation period. It should be noted that the proposed method does not depend on the length of observation period, as long as a sufficient number of vehicles is observed in each time slot. In more dense traffic, shorter observation windows can be used (e.g., 5 min) to provide sufficient data for analysis. A Doppler-based sensor was used as a reference device that measured the vehicles speed. No reference for the distance between the vehicles and the sensor was available.

In the first experiment, we tested speed estimation for individual vehicles with the proposed method. Figure 10 shows the results calculated from the position signal of each vehicle in the experiment M. It is clearly seen that the results are too inaccurate to obtain a reliable estimate of a single vehicle speed. There are two factors that contribute to the observed errors: noise present in the intensity signals and inaccuracy of estimation of vehicle’s distance from the sensor. Therefore, further experiments were focused on the time-averaged traffic speed estimation.

Earlier in the paper we stated an assumption that errors in distance and speed estimates, calculated for individual vehicles, are normally distributed. Figure 11 shows histograms of both types of errors, observed in the experiment M. The average error (mean ± standard deviation) is 0.35 ± 0.06 m for the distance to source estimation, and 2.28 ± 11.0 km/h for the speed estimation. Distribution of both error types is normal, as confirmed by the Shapiro-Wilk statistical test (distance: *W* = 0.98, *p* = 0.91; speed: *W* = 0.97, *p* = 0.5). Although the strength of the Shapiro-Wilk for such a small sample is low, the results indicate that there is no reason to reject the hypothesis that the population of estimation errors is normally distributed. Therefore, the assumption of the proposed method is confirmed.

Figure 12 and Figure 13 illustrate how the proposed method of time-averaged traffic speed estimation works. The following signals are shown: positions of the individual vehicles, the averaged position signal, and the position of a modelled point source, moving at a speed equal to the average speed of all vehicles (measured with the reference device). In the experiment M (Figure 12), the drivers attempted to move at a constant speed. This is reflected in position signals that are approximately linear, changes in the slope are mainly due to variations in distance. The averaged position signal is mostly aligned with the reference, although it deviates from it with the increasing distance from the sensor (but outside the analysis window used by the method). For the second experiment (L1; the remaining L experiments yielded similar results and they are not shown to avoid redundancy), much larger nonlinearity is observed in the position of individual vehicles. In this case, uncontrolled traffic was recorded, and the vehicles moved with varying speed and distance. The averaged position signal is also noticeably non-linear. Nevertheless, the linear regression applied to the average position provides a result that is reasonably consistent with the reference.

Table 1 shows the results of speed estimation in all experiments. The ground truth speed values were obtained from the reference devices. The results obtained using a simple approach of averaging individual speed estimates are included for comparison with the proposed method. The simple averaging method is quick, but it is only able to average errors in the computed speed estimates. With this approach, the average absolute error of speed estimation is 2.2 km/h (2.7%), which may be sufficient for the purpose of traffic monitoring. However, with the proposed method, which averages errors in the position signals used for speed estimation, instead of averaging errors in the final estimates, an improvement in speed estimation accuracy is obtained. The average error from all experiments is 1.2 km/h (1.44%), which is a reduction by 1 km/h (1.3 percentage points) compared with the simple averaging method. The increase in accuracy is especially notable in the experiment M (with controlled vehicle speed), where the error is reduced by 2.5 percentage points, and the obtained estimate is very close to the reference value (−0.3 km/h difference). In the L experiments (with uncontrolled vehicle speed), estimation errors are larger, but they are still lower than in the simple method, and they are at an acceptable level. It may also be observed that the number of observed vehicles is not a major factor affecting the estimation accuracy. It may be expected that a larger number of vehicles will result in a lower estimation error, because if a larger number of position signals is used, the averaged position signal becomes smoother and more linear. This can be observed by comparing Figure 12 (larger number of vehicles) with Figure 13 (smaller number of vehicles, the averaged position signal is less linear). However, the error in experiment L1 (8 vehicles) is the same as in the M experiment (31 vehicles), while errors in the L2 and L3 experiments (7 vehicles in both) differ significantly. The largest error was observed in the experiment L2. From the analysis of position signals, it may be concluded that the main factor that affects the estimation accuracy is variation in the vehicle speed (when the vehicle accelerates or brakes rapidly within the measurement area) and in the distance of vehicle’s trajectory from the sensor. It is expected that the estimation error is reduced if more vehicles are used in computations. However, the results of L experiments confirmed that a low number of vehicles (5–8) is sufficient to obtain an average speed estimate with a sufficient accuracy. Overall, the results confirm that the proposed method is valid and that it increases the speed estimation accuracy compared with the simple method that averages the estimates.

The results of our experiments validated the accuracy of the proposed algorithm by comparison of its results with data from state-of-art traffic monitoring sensors: a lidar and a radar. It is also interesting to compare our results with other approaches based on audio analysis. A review of state-of-art sensors revealed that there are currently no commercial devices capable of measuring traffic speed with acoustic sensors. There are a few acoustic sensors for vehicle detection and counting. Other published works on acoustic determination of vehicle speed focus on measuring speed of single vehicles and they often report large measurement errors. A probable explanation is that these methods treat vehicles as point sources, which (as we discussed in this paper) is not a valid assumption and it leads to large speed estimation errors. We compared the average speed error of 1.2 km/h obtained using our method with mean absolute error reported in several publications related to a wide range of acoustic-based traffic monitoring. Na et al. estimated the vehicle speed from measuring the time needed to pass through the detection zones in a multi-zone detector based on a microphone array. The reported average speed error from 940 vehicles was 16.58 km/h [17]. López-Valcarce et al. employed a maximum likelihood approach using two microphones. Only three vehicles were measured and the average error was 4 km/h [21]. Cevher et al. examined acoustic wave patterns from a single omnidirectional microphone. With a full profile method, they obtained average error of 2.97 km/h (from 10 vehicles), while two simplified methods provided errors of 13.39 km/h and 8.14 km/h [22]. Wu et al. examined acoustic patterns from smartphone audio and obtained average error of 2.17 km/h [34]. Göksu estimated the vehicle speed from on-board microphone and reported mean absolute error of 1.11 km/h [35]. Ishida et al. examined a map of sound arrival time difference using two microphones and obtained average speed error of 16.8 km/h for 12 vehicles, 9.38 km/h for 6 vehicles with motorcycles excluded [36]. We did not find any published works on using sound intensity analysis for traffic speed estimation which confirms that our approach is novel. Average speed error obtained in our experiments is lower than reported in other related publications (except for [35], where the error values are comparable).

## 4. Conclusions

In our previous publication [14], we presented a method of vehicle detection and counting by the analysis of sound intensity signals. In this paper, we extend this approach by proposing a novel approach to road traffic speed estimation, based on the analysis of sound intensity. Initially, we hoped to use this method to estimate the speed of each individual vehicle. However, this proved to be problematic, so we decided to develop a method that estimates an average traffic speed within an observation period, with the assumption that the averaging procedure reduces influence of various factors on the speed estimation accuracy. We identified three main sources of speed estimation errors. (1) The apparent sound source is not a point that is constant within a moving vehicle (a fact that is often neglected in related publications), also the source position depends on the analyzed frequency range. In our method, we diminish the influence of this factor by analyzing only a short time section when a vehicle is close to the sensor, and we perform analysis in three frequency bands. (2) Estimation of the distance between the movement path of a vehicle and the sensor is crucial for obtaining an accurate speed estimate. This proved to be the hardest problem, which we partially solved in the proposed method by analyzing the source elevation in two frequency bands, in a short time fragment when a vehicle is close to the sensor. (3) The signal-to-noise ratio in the calculated intensity and position signals is low, which results in errors in the calculated speed estimates. This problem may be diminished by averaging a number of speed estimates within the observation period, and this approach yields a satisfactory estimation accuracy. However, we proposed a method based on averaging the position signals obtained for individual vehicles and computing the speed estimate from the average position signal. The experiments proved that this method provides more accurate speed estimates than the simple averaging method.

An example of practical application of the proposed method is a traffic monitoring station in a smart city system. With this method, a vehicle count (in two opposite directions) and the average traffic speed can be calculated in the defined time slots (e.g., every 15 min). A network of such monitoring stations would be able to provide dynamic information on road traffic, suitable for urban traffic management. The advantage of the proposed method is that it is passive, it does not emit any signals, unlike the radars and lidars, so it does not contribute to signal pollution in the urban areas. It is also not susceptible to electromagnetic interference, which is the problem for radar-based sensors (as observed in the M experiment, in which the radar simply did not work). The proposed method is cost-effective, it does not require expensive equipment and high energy consumption, unlike lidar-based sensors. A low-cost setup consisting of six MEMS microphones, an I2S-USB interface and a Raspberry Pi microcomputer, which we used in our experiments, is sufficient to perform online analysis in a monitoring station. The sensor is also easy to install, and it does not need a specific orientation relative to the traffic direction (the sensor is omnidirectional), which radars and especially lidars require.

As is the case for each sensor-based, automated traffic monitoring method, the proposed approach also has its limitations. High level environmental noise may disturb the sound analysis. Very strong wind or a heavy rain with raindrops hitting the sensor enclosure may mask the signals emitted by vehicles. Such problems occur also in other sensors (e.g., a Doppler radar did not work during a heavy rainfall). There is also a problem of occlusion when multiple vehicles are observed by the sensor concurrently. The occlusion problem is common to most traffic monitoring sensors. Since the proposed method averages a number of vehicles, the occlusion is not critical for obtaining valid results. Another limitation of the proposed method is that it relies on the data provided by the vehicle detector. Occlusions are a common problem in traffic analysis. Partial occlusions are handled correctly by the algorithm, as long as a sufficiently long signal section around the zero azimuth is provided by the detector. Complete occlusions cannot be handled by the detector and such cases are excluded from the analysis. However, a reasonably small percentage of occlusions is not a problem for the speed estimation algorithm.

In the future work, the main possible area of improvement is estimation of the distance between the sensor and the sound source. Temporal resolution of the analysis may be increased, provided more data for successful speed estimation. We also intend to focus on improving the accuracy of speed estimation of individual vehicles. The method described in this paper will be implemented in a traffic monitoring station and we intend to perform a long-term, 24/7 analysis of road traffic in selected location. This test will provide more information on performance of the proposed method under different conditions.

## Figures and Tables

**Figure 1 sensors-21-05337-f001:**
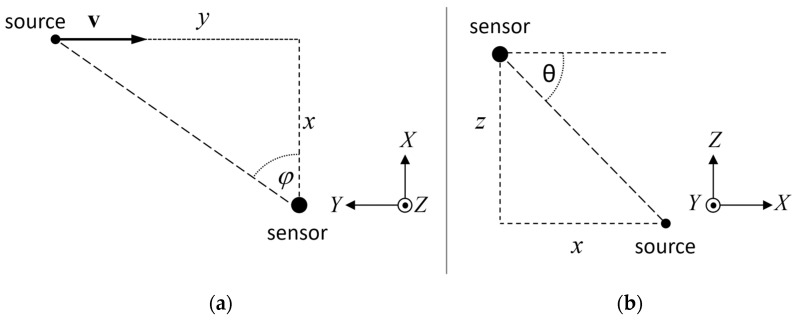
Orientation of the AVS coordinate system: (**a**) top view, (**b**) side view. The sensor position is (0, 0, 0), the sound source position is (*x*, *y*, *z*).

**Figure 2 sensors-21-05337-f002:**
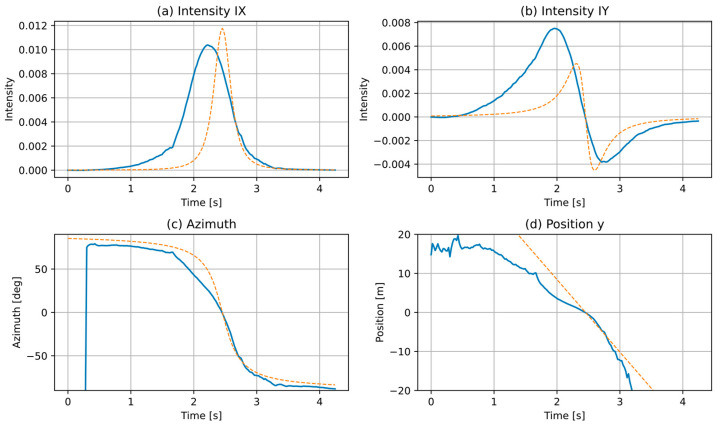
Comparison of signals recorded from a real road vehicle (solid line) with an ideal point source moving at the same speed (dashed line).

**Figure 3 sensors-21-05337-f003:**
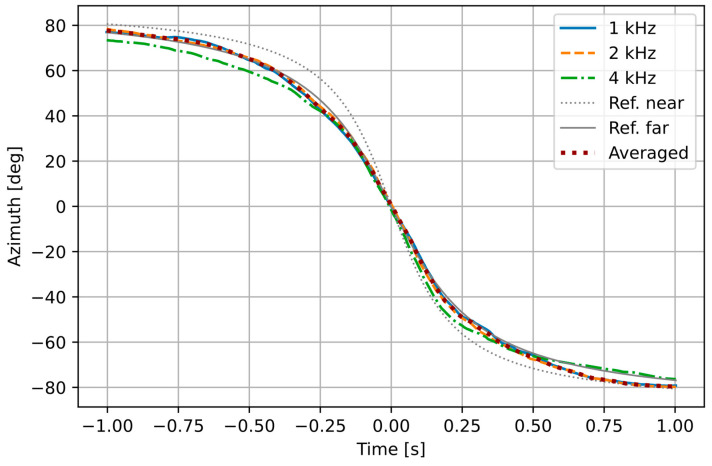
Azimuth measured by the AVS and computed by the model.

**Figure 4 sensors-21-05337-f004:**
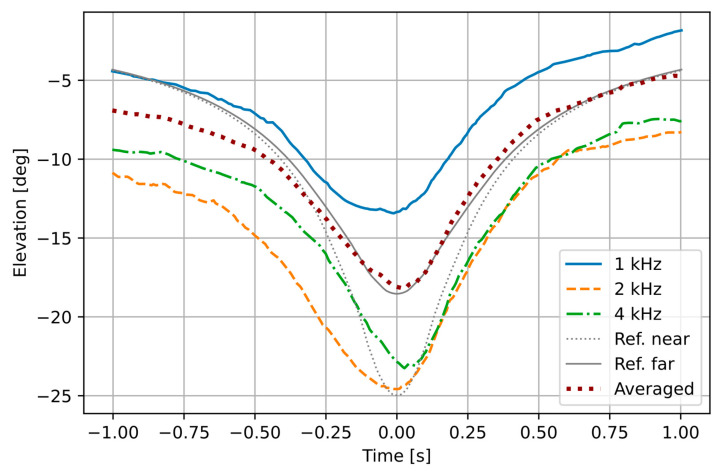
Elevation measured by the AVS and computed by the model.

**Figure 5 sensors-21-05337-f005:**
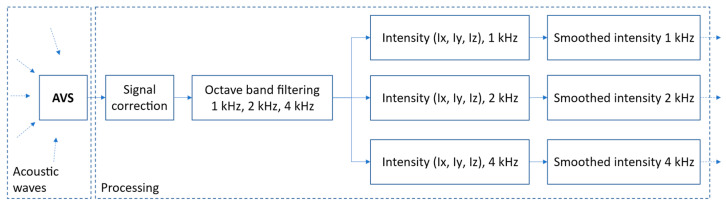
Block diagram of signal preprocessing and calculation of the smoothed intensity signals in frequency bands.

**Figure 6 sensors-21-05337-f006:**
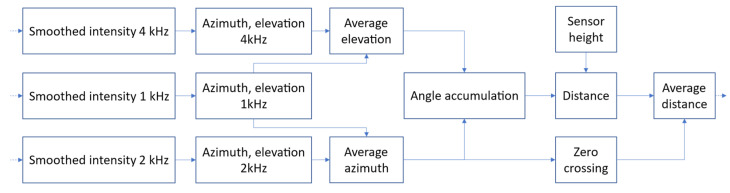
A block diagram of estimation of distance to the sound source.

**Figure 7 sensors-21-05337-f007:**
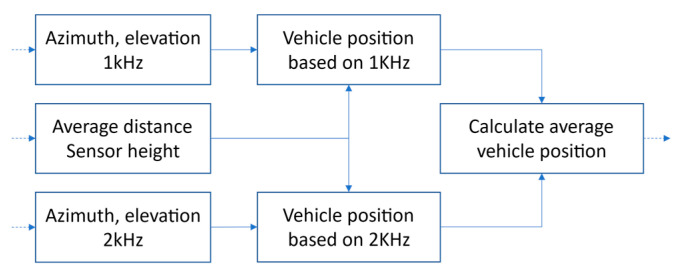
A block diagram of the average vehicle position estimation.

**Figure 8 sensors-21-05337-f008:**
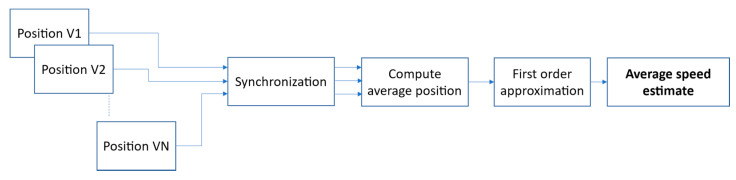
A block diagram of the average traffic speed estimation procedure.

**Figure 9 sensors-21-05337-f009:**
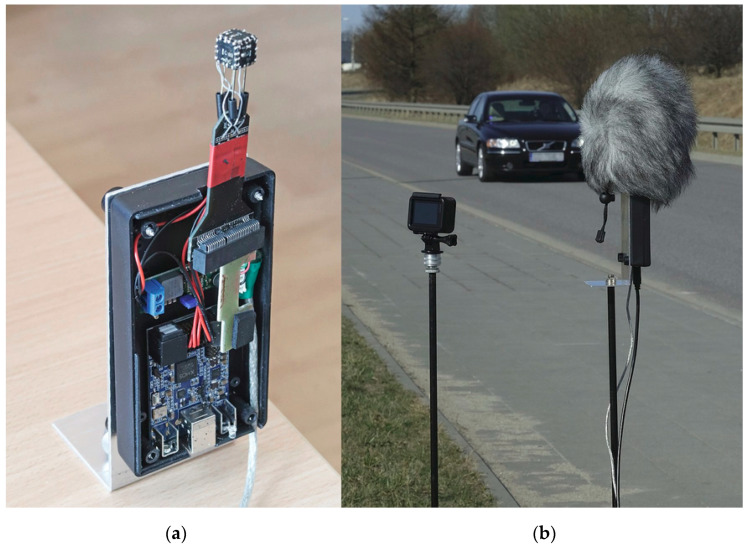
(**a**) The AVS used in the experiments. (**b**) The setup in the experiment M (AVS on the right, with a wind shield).

**Figure 10 sensors-21-05337-f010:**
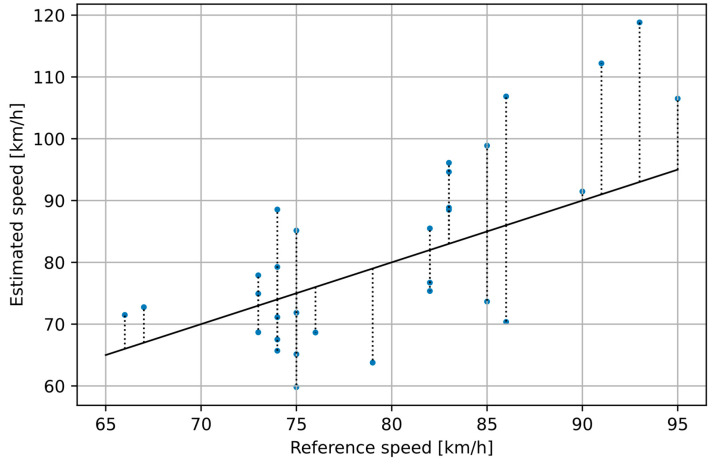
Estimated speed of individual vehicles vs. vehicle speed measured with the reference device.

**Figure 11 sensors-21-05337-f011:**
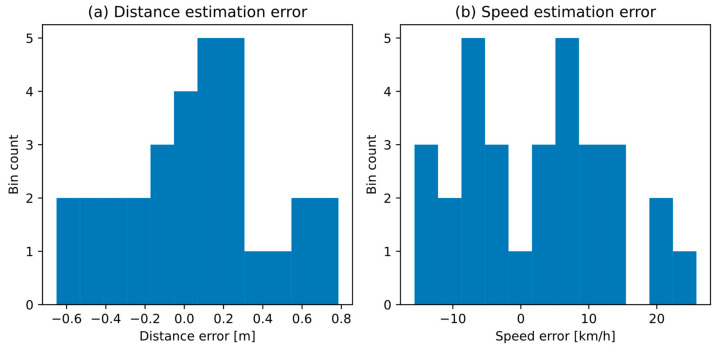
Histograms of estimation error for individual vehicles: (**a**) distance, (**b**) speed.

**Figure 12 sensors-21-05337-f012:**
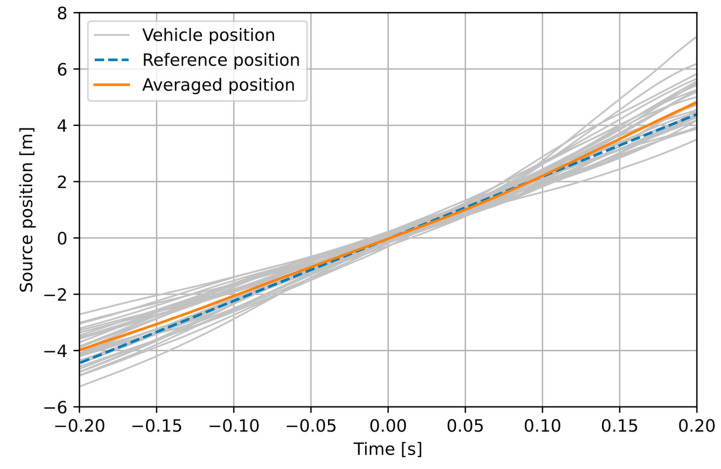
Position signals calculated in the experiment M.

**Figure 13 sensors-21-05337-f013:**
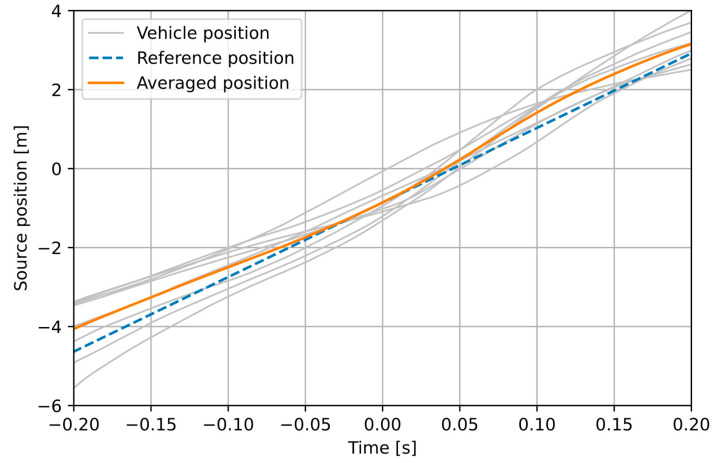
Position signals calculated in the experiment L1.

**Table 1 sensors-21-05337-t001:** Results of vehicle speed estimation with the proposed method.

Method	Parameter	M	L1	L2	L3	L4
Reference	Number of vehicles	31	8	7	5	7
	Average speed [km/h]	79.5	68.0	86.2	77.3	87.7
Simple averaging	Average speed [km/h]	81.8	67.3	83.1	75.0	85.3
	Estimation error [km/h]	2.3	−0.7	−3.1	−2.3	−2.4
	Estimation error [%]	2.9	1.0	3.6	3.0	2.8
Proposed method	Average speed [km/h]	79.2	67.7	83.4	76.1	89.1
	Estimation error [km/h]	−0.3	−0.3	−2.8	−1.2	1.4
	Estimation error [%]	0.4	0.4	3.2	1.6	1.6

## Data Availability

The data included in this study are available upon request by contact with the corresponding author.

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
