# Peer review of "Estimation of Average Speed of Road Vehicles by Sound Intensity Analysis"

_sensors, 2021, doi:10.3390/s21165337_

Round 1

Reviewer 1 Report

The authors propose an acoustic-sensor-based traffic monitoring system exploiting sound intensity. The proposed system has low costs and simplifies implementation.

Strengths:

+ The paper is very well written.

+ The idea of utilizing the sound intensity of an acoustic sensor for implementing a traffic monitoring system seems novel.

+ The algorithm designed to reduce the impact of noise is interesting.

+ The experiments are performed in both controlled and uncontrolled environments. 

+ The experiments are performed in a real-world environment. 

Weaknesses:

- The motivation of the proposed approach can be explained better. The advantages of using acoustic sensors for traffic monitoring compared to other types of sensor should be more clearly described. 

- When vehicles are overlapped, it would be very difficult, if not possible, to analyze the sound signals for traffic monitoring. The authors should provide details on how the proposed system handles detecting and estimating the speed of overlapped vehicles. 

- While the proposed approach attempts to address the noise issue, there are numerous other sources of noise, especially in urban roadways. It is not clear how the proposed approach could generalize and reduce the impact of noise from various sources.

- It is mentioned in the paper that the due to the high level of external electromagnetic interference, it was impossible to use the radar sensor as a reference device. The authors should mention more details about this interference.

- What is the state-of-the-art vehicle speed estimation system? How does the proposed work perform compared with the state-of-the-art system? Why should the proposed system be used compared with other solutions? The proposed system is compared only with a reference system. What is the currently-known best acoustic-sensor-based traffic monitoring system and what is the performance of the state-of-the-art approach?

- All graphs look blurry.

Reviewer 2 Report

The authors of the paper „Estimation od average speed of road vehicles by sound intensity analysis” present the novel approach based on acoustic vector sensors data analysis for the traffic speed estimation. The paper is overall well written and easy to follow. I would suggest, however, that the authors resolve some issues prior to the paper being published.

The main remarks are as follows:

  1. In experiment M there should be 4*7*2 =56 runs, but only 31 were used in experiments. It should be explained why not all runs were used.
  2. In the second experiment the choice of the time slots should be better explained. Why only recordings of isolated vehicles were selected?
  3. The authors claim that the occlusion is not critical for obtaining valid results. „...the method works as well for more dense traffic, in shorter observation windows (e.g., 5 minutes), as confirmed in our experiments.” - the results of these experiments should be given in the paper to prove it.
  4. „We aim to show that a small number of vehicles is sufficient to obtain an accurate speed estimate.” - is it really accurate, since even in the experiment with controlled speed „the results are too inaccurate to obtain a reliable estimate of a single vehicle speed”?
  5. It is not clear why in experiment M the authors do not use other sources of information about the vehicle’s speed. The readings from OBD about the actual speed of the vehicle or indications of the speedometer can be used and compared with the information from the lidar. It would be good if the authors provided more information about the lidar used in the research, in particular about its accuracy.
  6. Are the estimation errors in Table 1 the average errors for individual runs? If so, the average absolute error of speed estimation should be calculated as the average absolute error of speed estimation for each run, not as the average of averages.
  7. A bit alarming is the fact that the authors use the Shapiro-Wilk test using only 31 samples to confirm the assumption that distance and speed errors are normally distributed. When the distribution is similar to the normal distribution, the effect size is small (the power of the test depends of the effect size) and a large sample size is required. Moreover, the authors admit that the histogram of speed estimation error is bimodal, so it can be concluded that it cannot be normally distributed. If the distribution is bimodal it would be worth analyzing these two groups separately.
  8. More information about recordings should be given - at least the weather conditions during recordings and the type of the road (is it city road or suburban road?). Five observed vehicles per hour seem to be little, even at night.
  9. Some sentences should be proven or the reference should be given. In particular, please justify the following statements:
  10. „Obtaining more detailed data, such as determining the vehicle type or the road surface state, is also possible by further analysis of the AVS signals”
  11. „Lower frequency bands were rejected because they usually contain a high level of noise (e.g., wind), and higher frequency bands have too low signal-to-noise ratio.”
  12. „It may be expected that a larger number of vehicles will result in lower estimation error.”
  13. The origin of equation 13 is unknown - it should be better explained how such a formula was obtained.
  14. Some figures are of very low quality in the revised version of the paper (especially figures 2, 5, 6, 7, 8, but also figs. 12 and 13). Please, make sure that in the final paper they are in vector resolution.
  15. Please reformulate the following sentences to make them easier to understand:
  16. „At the same time, attenuation of sound intensity with increasing distance from the source is larger for higher frequencies, so the relation between sound intensity measured in 1 kHz and 4 kHz bands will be different for a vehicle close to the sensor and for the same vehicle at a lager distance.”
  17. „The sensor is also easy to install, and it is omnidirectional, so it does not have to be aimed that the traffic, which radars and especially lidars require.”
  18. Minor editing mistakes, e.g.
  19. Bargagli -> Barbagli (l. 95)
  20. Section -> section (l. 119)
  21. drection->direction (l.124)
  22. area. [27,28]. -> area [27,28]. (l.127)
  23. b -> x (Eq. 6)
  24. lager -> larger (l.230)
  25. uniformly samples -> uniformly sampled (?) (l.287)
  26. c.a. -> ca (l.354, it should be the same as the previously used form)
  27. 31 vehicles -> 31 runs (?) (l.381)
  28. References 26-31 should be corrected.
  29. All figures’ captions should be placed under the figures, on the same page.
  30. In Table 1 horizontal lines between the methods should be added to increase the readability.

Reviewer 3 Report

In this paper, the authors proposed a method to estimate average speed of road vehicles in the observation period, using a passive acoustic vector sensor. However, there are some problems as follow:

1.In what specific scenarios can the method used in this paper be used for velocity estimation? What is the use of this result? What is the accuracy of the data? These questions need to be made clear in the introduction.

2.The advantages and disadvantages of proposed a method compared with radar and other traditional schemes are presented in the introduction and conclusion of this paper, but there is no clear analysis of the points, it is best to be listed in the introduction.

3.Is the letter b in Formula 6 a typo? It should be x.

4.In the azimuth result in Figure 2, there is a huge difference between the real value and the measured value near the abscissa of 0. What is the reason for this difference? The result of position y is also quite different, and sufficient analysis should be given.

5.In section 2.4, the analysis of the relationship between the three frequency bands and sound intensity and how formula 10 was calculated need to be illustrated by figures.

6.The authors need to indicate whether the positioning of the sensors in section 2.4 is their own design or whether there are other references or standards. If it is their own design, they need to explain the motivation.

Round 2

Reviewer 1 Report

Most of my comments have been addressed adequately. However, it is still not clear how the proposed work performs compared with the state-of-the-art "acoustic sensor"-based vehicle speed estimation algorithm. Here, the state-of-the-art approach does not mean the device that generates the ground-truth data. For example, here are some papers about acoustic-sensor-based vehicle speed estimation. 

Wu, Yue, et al. "HDSpeed: Hybrid Detection of Vehicle Speed via Acoustic Sensing on Smartphones." IEEE Transactions on Mobile Computing (2020).

Cevher, Volkan, Rama Chellappa, and James H. McClellan. "Vehicle speed estimation using acoustic wave patterns." IEEE Transactions on signal processing 57.1 (2008): 30-47.

The authors are suggested to add performance comparison with existing algorithms for acoustic-sensor-based vehicle speed estimation.

Author Response

The authors thank the Reviewer for analysis of the manuscript. In response to  Reviewer’s suggestion, we added a paragraph at the end of Section 4 in which we compare the accuracy of our method with several published works on the related topic. It should be noted that these publications focused on measuring speed of individual vehicles, and although these methods belong to a wide range of acoustic approaches, none of them is based on analysis of sound intensity, which we use in our method. We compared the average speed error from our experiments with mean absolute error values reported by other researchers.

Regarding the state-of-art sensors used for traffic monitoring: to our knowledge, there are currently no commercial, certified acoustic sensors that are capable of measuring the vehicle speed. There are a few acoustic sensors that detect and count the vehicles, but they do not measure speed. An example of such a device is SAS-1 Passive Acoustic Detector by SmarTek Systems (http://www.smarteksys.com/SASpad.html). Therefore, to obtain a reference data for validation of our algorithm, we had to rely on the commonly used sensors: a lidar and a radar.

Reviewer 2 Report

All my comments were taken into account, and appropriate changes were made in the manuscript. It can be published in the Sensors in present form.

Author Response

We thank the Reviewer for reviewing our manuscript.

Reviewer 3 Report

No further comments.

Author Response

(The authors gave the same response as above.)
